# Clinicopathological and Prognostic Characteristics of RAD51 in Colorectal Cancer

**DOI:** 10.3390/medicina56020048

**Published:** 2020-01-21

**Authors:** Jae-Ho Lee, An-Na Bae, Soo-Jung Jung

**Affiliations:** Department of Anatomy, Keimyung University School of Medicine, Daegu 42061, Korea; anato82@dsmc.or.kr (J.-H.L.); sand31@hanmail.net (A.-N.B.)

**Keywords:** RAD51, colorectal cancer, double-stand breaks, cancer prognosis, clinical analysis

## Abstract

*Background and Objectives*: RAD51 plays an essential role in DNA repair via homologous recombination. RAD51 facilitates strand transfer between interrupted sequences and their undamaged homologies. Therefore, we studied the RAD51 mRNA expression levels in colorectal cancer (CRC), and evaluated the clinicopathological and prognostic significance of RAD51. *Materials and Methods*: The RAD51 expression was examined in 48 CRCs and paired adjacent non-tumor tissues. We further evaluated the survival to determine the prognostic value of RAD51 in our CRC and The Cancer Genome Atlas (TCGA) data. *Results*: We confirmed that the RAD51 expression in tumor tissues, compared with that of paired non-tumor tissues, was upregulated 2.5-fold. Additionally, the RAD51 expression was significantly associated with the T stage (*p* = 0.027). According to a higher T stage, the RAD51 expression showed an increasing trend. However, the RAD51 expression did not show a prognostic value statistically. *Conclusions*: We confirmed that RAD51 was upregulated in tumors and was significantly associated with the T stage. Although there was no statistically significant prognostic value found in our samples and TGCA data, our study will provide new insight for RAD51 in CRC.

## 1. Introduction

Colorectal cancer (CRC) is one of the main reasons of cancer-related deaths worldwide, and is characterized by a heterogeneous tumor harboring different cell populations with distinct properties, contributing to the disease complexity [1]. Moreover, CRC is one of the most common cancers, with an increasing frequency, especially in Korea. CRC represents a universal model for cancer initiation and progression through well-defined histological stages [2]. Genetic changes in CRC have been broadly continued. Although many authors have studied the epidemiology of CRC for a long time [3,4,5,6], little is known about the latest tendencies of the demographic features and genetic alterations of CRC.

The new target therapy induced dramatic improvements in cancer treatment over the recent years [7]. Double-stand breaks (DSBs) cause severe damage and the homologues recombination (HR) pathway is an extensively regulated process for DSBs repair. Particularly, with regard to DNA repair, poly (ADP-ribose) polymerase inhibitors show promise as a powerful therapeutic tool, especially in the management of tumors deficient in HR [8]. RAD51 has an essential role in DSBs repair through HR [9]; after specific DNA damage, RAD51 localizes to nuclear foci for DNA repair. As part of HR, RAD51 helps strand transfer between interrupted sequences and their undamaged homologies [10]. Many studies have generally suggested that the RAD51 expression increased cellular resistance to chemotherapy and radiotherapy, like crosslinking agents or topoisomerase inhibitors [11,12,13]. However, this remains controversial, as contradictory results were shown in some cancers. A similar study also revealed that lung cancer patients harboring a strong Rad51 expression had a significantly poorer survival than those without a Rad51 expression [14]. Substantial further study is needed to clarify its underlying mechanisms.

In this study, we explored the RAD51 mRNA expression in CRC, and evaluated the clinicopathological and prognostic characteristics of RAD51; we suggest that RAD51 may serve as an important candidate for CRC treatment.

## 2. Materials and Methods

### 2.1. Patients and Tissue Samples

Altogether, 48 patients diagnosed with CRC were included in the present study. Colorectal adenocarcinomas and paired non-cancerous samples were obtained from patients undergoing surgery at Dongsan Medical Center (Daegu, Korea), between June 2008 and November 2010. The tissues were proximately frozen in liquid nitrogen and stored at −80 °C until RNA isolations were performed. The tissues were provided from Keimyung Human Bio-Resource Bank, Korea. The purpose of the study was explained to all of the patients, and informed consent was obtained from each study participant. The study was conducted in accordance with the Declaration of Helsinki, and the protocol was approved by the Institutional Review Board of Keimyung University Dongsan Medical Center (approval no. 12–41). The exclusion criteria included preoperative chemoradiotherapy, a previous history of surgical resection for CRCs, death within 30 postoperative days, and evidence of hereditary non-polyposis colorectal cancer (Amsterdam criteria) or familial adenomatous polyposis. The classification system of the American Joint Committee on Cancer (seveth edition) was used to determine the pathological tumor depth, the number of metastasized lymph nodes, and the cancer stage. Recurrence was defined as the presence of a histologically and/or radiologically confirmed tumor. Clinicopathological data, including age, pathologic tumor and nodal stage, preoperative carcinoembryonic antigen (CEA), lymphovascular invasion, perineural invasion, tumor differentiation, and survival data were obtained. An experienced pathologist reviewed all of the cases and their clinicopathological characteristics were presented.

### 2.2. RNA Isolation and mRNA Expression Analysis

The total cellular RNA was extracted from the tissues using a QIAzol lysis reagent (Qiagen, CA, USA), according to the manufacturer’s protocol. A NanoDrop ND-1000 spectrophotometer (Thermo Fisher Scientific, Waltham, MA, USA) was used to determine the quantity and quality of the isolated total cellular RNA. Reverse-transcription reactions were performed using a High-Capacity cDNA Reverse Transcription Kit (Applied Biosystems, Foster City, CA, USA). The expression levels of the RAD51 mRNA were determined through quantitative reverse-transcription polymerase chain reaction (qRT-PCR) using a Power SYBR Green master mix (Toyobo, Osaka, Japan). qRT-PCR analyses were performed on the CFX Connect™ Real-Time PCR Detection System (Bio-Rad, Hercules, CA, USA), using the following primers: forward-RAD51 (5′-TCTCTGGCAGTGATGTCCTGGA-3′), reverse-RAD51 (5′-TAAAGGGCGGTGGCACTGTCTA-3′), forward-GAPDH (5′-ACCCACACTGTGCCCATCTAC-3′), and reverse-GAPDH (5′-TCGGEGAGGATCTTCATGAGG-3′). To avoid inter-plate variations, cycle threshold (Ct) values were calculated using the same threshold cut-off values for each assay. The relative expression was calculated using the ∆Ct method, and each experiment was performed in duplicate.

### 2.3. The Cancer Genome Atlas (TCGA) Data Analysis

We used public TCGA (http://cancergenome.nih.gov/) data. In total, 598 CRC tumors with clinical data (data of download: October 2019) were profiled for the survival analysis. Survival was defined as the time interval from surgery until the date of death.

### 2.4. Statistical Analysis

Chi-square, Fisher’s exact test, and simple correlation analyses were used to analyze the association between the variables. Survival curves, constructed using the univariate Kaplan–Meier estimators, were compared using the log-rank test. Overall survival (OS) was defined as the time between diagnosis and mortality. Disease-free survival (DFS) was defined as the time between diagnosis, and disease recurrence or the development of distant metastasis. The correlations between telomere length and mRNA expression and clinicopathological parameters were assessed by Pearson’s correlation coefficient analysis. A *p*-value of <0.05 denoted significance in all of the statistical analyses performed in the study.

## 3. Results

### 3.1. Patient and Tumor Characteristics

The RAD51 mRNA expression was successfully analyzed in all 48 cases of CRC. The average ΔCt values of RAD51 mRNA expression in CRC tissue was 3.72 ± 2.7, and those observed in paired adjacent non-tumor tissues was 5.5 ± 2.34. Therefore, the RAD51 expression in cancer, compared to the paired non-tumor tissue, was upregulated by 2.55-fold (*p* < 0.0001). To further explore the association between the RAD51 expression and the clinicopathological variables of CRC, patients were divided into two groups, according to the average values of the tumor/non-tumor ratio. RAD51 mRNA expression was categorized as negative in 39 (81.2%) patients and positive in nine (18.8%) patients. The patient and tumor characteristics of the study cohort separated by RAD51 mRNA status are presented in Table 1.

The demographic characteristics between the two groups were similar for age, body mass index (BMI), cancer stage, CEA level, differentiation, lymphovascular invasion, and perineural invasion. The RAD51 mRNA expression was significantly associated with the T stage (*p* = 0.027), and other variables did not have any association with RAD51. A quantitative correlation analysis also showed that the RAD51 expression did not have a correlation to the age, BMI, and CEA level (Table 2).

### 3.2. Survival Analyses According to the Prognostic Model

We next assessed survival to determine the prognostic value of RAD51 expression in CRC. The median follow-up of patients for survival analysis was 67.76 months (range of 2–102 months). Survival results by the Kaplan–Meier analysis showed that the RAD51 expression was not associated with the overall survival (*p* = 0.408; Figure 1A) and disease free survival (*p* = 0.601; Figure 1B) in CRC.

When separating according to different variables, the RAD51 status did not appear to confer any statistically significant prognostic value. Furthermore, the RAD51 mRNA expression data were downloaded from the TCGA data portal to analyze the prognostic value of the RAD51 expression; the overall survival analysis of RAD51 expression did not show prognostic significance in colorectal cancer (*p* = 0.496; Figure 2A), colon cancer (*p* = 0.821; Figure 2B), and rectal cancer (*p* = 0.392; Figure 2C) in the TCGA data.

## 4. Discussion

The *Rad51* gene is a homolog of *RecA* in *Escherichia coli* [15], and is associated with DNA repair and recombination [16,17]. RAD51 binds to DNA and exhibits a DNA-dependent adenosine triphosphatase (ATPase) activity. It produces helical nucleoprotein filaments inducing homologous pairing and strand exchange between DNA molecules [18,19,20]. Mutations in Rad51 led to a decrease in DNA recombination, hypersensitivity to ionizing radiation and methylmethanesulfonate, and a lack in DSBs repair [21]. Previous studies showed that RAD51 is a main factor in HR in the DNA repair system [22,23].

The regulation of HR contains many protein interactions that are powerfully dependent on post-translational modification. Also, crucial mediator proteins of HR are subject to phosphorylation by specific kinases, thereby controlling some stages of this process (e.g., SUMOylation) [24]. Significantly, because RAD51 contributes to genome maintenance, and as it interacts with the tumor protectors such as BRCA1, BRCA2 and p53, it is thus supposed that RAD51 may take part in tumor inhibition [25]. So as to better comprehend the regulation of HR, the detail mechanism experiment will be needed to identify the interaction network of RAD51. RAD51 overexpression in cancer has been usually reported, and has induced poor prognosis, particularly in CRC [26]. Moreover, microarray analysis showed an increased RAD51 expression in pancreatic carcinoma [27]. Furthermore, this overexpression was shown in 66% of pancreatic cancer, and functional analysis suggested that RAD51 overexpression enhanced the cell survival after DSBs damage [28]. Rad51 may also operate other intracellular pathways, including p21, p53, and Bcl-2, causing the repair of damaged DNA or further amplifying the damaged recombinant DNA [29].

In present study, we explored the influence of RAD51 mRNA expression in 48 cases of CRC. We confirmed that RAD51 mRNA expression in tumor tissues was upregulated 2.5-fold when compared to paired adjacent non-tumor tissue. The most important data is that the RAD51 mRNA expression was significantly associated with T stage (*p* = 0.027); the RAD51 mRNA expression showed an increasing trend along with higher T stages. As it was thought that the T stage was indicative of cancer prognosis, RAD51 mRNA may be regarded as a marker of prognostic value.

Next, we evaluated the survival data to confirm the prognostic value of RAD51 mRNA in CRC. The RAD51 mRNA expression showed no prognostic value in our CRC samples, and the same result was derived from the TCGA data. However, Tennstedt et al. reported that the RAD51 expression was an independent indicator for CRC progression significantly, as were the tumor stage and nodal status. CRC patients with z RAD51 overexpression showed a median survival time, however, those with a low or no RAD51 expression showed poorer survival results [7]. Also, Qiao et al. introduced the RAD51 expression as an independent prognostic marker in lung cancer, with increased levels of RAD51 protein in tumors predicting poorer survival result for the individual patients [14]. Moreover, a high expression of RAD51 may be a significant prognostic marker in breast cancer, glioblastoma, and esophageal squamous cell carcinoma [23,30,31]. These results demonstrated that the high expression of RAD51 might result in a worse survival outcome by stimulating immortal cells [14]. Therefore, further studies on RAD51 expression in a variety of cancers are necessary. To clarify the function of the RAD51, more study about its protein expression level may be needed. The correlation between RAD51 mRNA and the protein expression levels should be also identified.

## 5. Conclusions

To conclude, our study analyzed the RAD51 mRNA expression levels in CRC, and evaluated the clinicopathological and prognostic potential of RAD51. Even though our data did not reveal any statistically significant prognostic value, RAD51 may serve as a valuable candidate for developing novel therapies for CRC treatment.

## Figures and Tables

**Figure 1 medicina-56-00048-f001:**
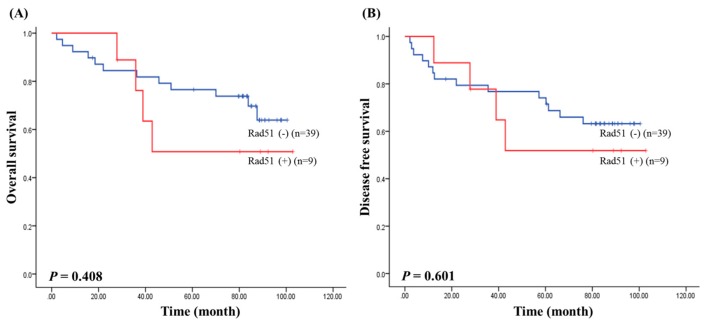
Survival analysis in colorectal cancer (CRC). (**A**) The relationship between overall survival and RAD51 expression; (**B**) the relationship between disease free survival and RAD51 expression.

**Figure 2 medicina-56-00048-f002:**
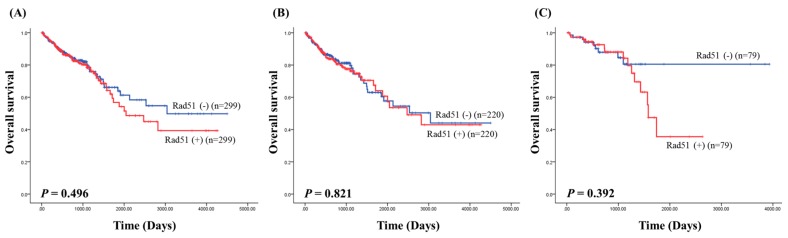
Overall survival analysis of RAD51 expression in The Cancer Genome Atlas (TCGA) data. (**A**) Colorectal cancer; (**B**) colon cancer; (**C**) rectal cancer.

**Table 1 medicina-56-00048-t001:** Characteristics of patients with colorectal cancer according to RAD51 mRNA expression. BMI—body mass index; CEA—carcinoembryonic antigen.

	RAD51 Expression		*p*
	(−)	(+)	
Total	39 (81.2)	9 (18.8)	
Age			0.741
≤60	15 (78.9)	4 (21.1)	
>60	24 (82.8)	5 (17.2)	
BMI			0.517
≤25	26 (78.8)	7 (21.2)	
>25	13 (86.7)	2 (13.3)	
T stage			0.027
T 1	0 (0.0)	2 (100.0)	
T 2	5 (83.3)	1 (16.7)	
T 3	26 (83.9)	5 (16.1)	
T 4	8 (88.9)	1(11.1)	
N stage			0.957
N 0	23 (82.1)	5 (17.9)	
N 1	9 (81.8)	2 (18.2)	
N 2	7 (77.8)	2 (22.2)	
CEA (ng/mL)			0.716
≤5	28 (80.0)	7 (20.0)	
>5	11 (84.6)	2 (15.4)	
Differentiation			0.111
Well-differentiated	1 (50.0)	1 (50.0)	
Moderately differentiated	36 (85.7)	6 (14.3)	
Poorly differentiated	2 (50.0)	2 (50.0)	
Lymphovascular invasion			0.348
(−)	15 (75.0)	5 (25.0)	
(+)	24 (85.7)	4 (14.3)	
Perineural invasion			0.091
(−)	23 (74.2)	8 (25.8)	
(+)	16 (94.1)	1 (5.9)	

**Table 2 medicina-56-00048-t002:** Correlation between the RAD51 mRNA expression and the clinical parameters in patients with colorectal cancer (CRC).

	Age	BMI	CEA	RAD51
**Age**	R	1	0.032	0.212	0.165
P		0.811	0.103	0.262
**BMI**	R	0.032	1	0.087	−0.046
P	0.811		0.510	0.756
**CEA**	R	0.212	0.087	1	0.041
P	0.103	0.510		0.779
**RAD51**	R	0.165	−0.046	0.041	1
P	0.262	0.756	0.779

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
