# Peer review of "Clinicopathological and Prognostic Characteristics of RAD51 in Colorectal Cancer"

_1010-660X, 2020, doi:10.3390/medicina56020048_

Round 1

Reviewer 1 Report

The authors have used qRTPCR to demonstrate that increased RAD51 levels correlate with reduced overall and progression -free survival of CRC patients. The manuscript is well written and data are presented clearly. However, the sample size for is study is 48 samples which is low. Additionally, it is well known that post-translational modifications of RAD51 are implicated in cancer pathogenesis. The authors should address this point in more details in the discussion section. It also important to know how the changes in the mRNA levels correlate with actual protein levels. however the latter might be beyond the scope of this study. 
Overall, the work is relevant to merit publication. 

Author Response

First of all we would like to thank the referees and editor for their time in reviewing our manuscript. Their constructive comments helped us to improve our work. We submit the revised version for your consideration. Please reconsider below a detailed response to your comments. We hope that our revised manuscript will be accepted for publication for MEDICINA

The authors have used qRTPCR to demonstrate that increased RAD51 levels correlate with reduced overall and progression-free survival of CRC patients. The manuscript is well written and data are presented clearly.

> Thank you for your kind review.

However, the sample size for is study is 48 samples which is low.

> It it very hard to gather good quality of RNA samples. We will add more samples in further study. 

Additionally, it is well known that post-translational modifications of RAD51 are implicated in cancer pathogenesis. The authors should address this point in more details in the discussion section. It also important to know how the changes in the mRNA levels correlate with actual protein levels.

> This decription was added in Discussion

however the latter might be beyond the scope of this study. 
Overall, the work is relevant to merit publication. 

> Thank you for your kind review

Reviewer 2 Report

This is a cohort study on mRNA expression level of RAD51 during different stages in colorectal cancer patients. Authors indicated the results support that RAD51 was upregulated in tumors and significantly associated with T stage. I do have few concerns to agree this manuscript to be published in Medicina journal.

The baseline of patents and inclusion criteria of patents are not provided.  The RAD51 not only cell cycle-regulated, but also it is could be affected by a lot factors, including treatment of patents (like hypoxic exposure, cisplatin treatment). The data provided here are considered to be selection bias. According to the data, authors should provide more detailed discuss, like involved pathway, potential mechanism on the T-stage correlation  and comparison between published data.

Author Response

First of all we would like to thank the referees and editor for their time in reviewing our manuscript. Their constructive comments helped us to improve our work. We submit the revised version for your consideration. Please reconsider below a detailed response to your comments. We hope that our revised manuscript will be accepted for publication for MEDICINA.

The baseline of patents and inclusion criteria of patents are not provided.  The RAD51 not only cell cycle-regulated, but also it is could be affected by a lot factors, including treatment of patents (like hypoxic exposure, cisplatin treatment). The data provided here are considered to be selection bias.

>patients criteria was added in Materials and Methods

According to the data, authors should provide more detailed discuss, like involved pathway, potential mechanism on the T-stage correlation  and comparison between published data.

> This decription was added in Discussion. We found previous results similar with our data and its potential mechanism was suggested.  

Round 2

Reviewer 2 Report

Thanks for authors response. I still feel the information about patents are not quite enough, like the treatments. Despite this, the overall data is still worth to be published and to be a small cohort references. 

In table 1, the number in Age section seems not correct. 

Author Response

First of all we would like to thank the referees and editor for their time in reviewing our manuscript. Their constructive comments helped us to improve our work. We submit the revised version for your consideration. Please reconsider below a detailed response to your comments. We hope that our revised manuscript will be accepted for publication for MEDICINA.

I still feel the information about patents are not quite enough, like the treatments. Despite this, the overall data is still worth to be published and to be a small cohort references. 

> Thank you for your kind review

In table 1, the number in Age section seems not correct.

> We have corrected the number in Age section.